# Regulation of B Cell Responses in SLE by Three Classes of Interferons

**DOI:** 10.3390/ijms221910464

**Published:** 2021-09-28

**Authors:** Phillip P. Domeier, Ziaur S. M. Rahman

**Affiliations:** 1Immunology Program, Benaroya Research Institute, Seattle, WA 98101, USA; pdomeier@benaroyaresearch.org; 2Department of Microbiology and Immunology, Pennsylvania State University College of Medicine, Hershey, PA 17033, USA

**Keywords:** interferons, B cells, systemic lupus erythematosus

## Abstract

There are three classes of interferons (type 1, 2, and 3) that can contribute to the development and maintenance of various autoimmune diseases, including systemic lupus erythematosus (SLE). Each class of interferons promotes the generation of autoreactive B cells and SLE-associated autoantibodies by distinct signaling mechanisms. SLE patients treated with various type 1 interferon-blocking biologics have diverse outcomes, suggesting that additional environmental and genetic factors may dictate how these cytokines contribute to the development of autoreactive B cells and SLE. Understanding how each class of interferons controls B cell responses in SLE is necessary for developing optimized B cell- and interferon-targeted therapeutics. In this review, we will discuss how each class of interferons differentially promotes the loss of peripheral B cell tolerance and leads to the development of autoreactive B cells, autoantibodies, and SLE.

## 1. Introduction

An interplay of immune dysregulation and environmental triggers causes autoimmunity in individuals with genetic susceptibility to various autoimmune diseases. B cell-mediated autoimmune diseases are characterized by autoantibodies that target healthy tissues and cause pathology. In systemic lupus erythematosus (SLE), patients develop autoantibodies that are reactive to nuclear antigens (called anti-nuclear antibodies, ANAs), including DNA (deoxyribonucleic acid), nucleosomes, ribonucleoproteins, and RNA (ribonucleic acid). Due to the abundance of nuclear autoantigens throughout the body that can be targeted by ANAs, SLE patients often develop a complex disease affecting multiple organ systems. Nuclear autoantigens can be released from dying cells by environmental insults, which then lead to the induction of pro-inflammatory cytokines, including interferons. Normally, interferon-mediated inflammatory responses are required for immune protection against various pathogens, but an elevated interferon response is also a hallmark of SLE [1].

### 1.1. Types of Interferons

Interferons are organized into three classes by target receptor binding, and these interferon receptors elicit distinct signaling cascades in specific immune cell populations (Table 1). Environmental factors, including ultraviolet light exposure, pollutants, microbiota dysbiosis, and viral infections, activate pattern recognition receptors (PRRs) or cell stress responses to induce the release of interferons [2,3,4,5]. Several endosomal and cytosolic PRRs, including Toll-like receptors (TLRs), NOD-like receptors (NLRs), and RIG-I-like receptors (RLRs) induce interferon production after binding to pathogen-associated molecular patterns (PAMPs) [6]. In addition to activation by PAMPs, PRRs can also promote autoreactivity after being activated by cell debris, circulating immune complexes, and/or damaged/dying cells [7]. The cytosolic DNA sensor cyclic GMP-AMP synthase- Stimulator of interferon genes (cGAS-STING) also promotes interferon responses both during anti-pathogen and autoimmune conditions [8,9]. Different subsets of B cells express varied levels of receptors to detect and mediate signals from all three types of interferons, but how these different classes of interferons independently and cooperatively promote autoreactive B cell responses in SLE continues to be studied.

Type 1 interferons (T1IFN) in mammals include interferon alpha (α), beta (β), omega (ω), delta (δ), nu (ν), kappa (k), tau (τ), and epsilon (ε) [10,11]. Humans have 14 interferon α genes (including 2 pseudogenes and 1 duplicate gene), 1 interferon β, 1 interferon ω, 1 interferon δ (a pseudogene), 1 interferon ν (a pseudogene), and 1 interferon ε. However, mice have 13 interferon α genes (with 1 pseudogene), 1 interferon β, 1 interferon κ, 1 interferon ε, 7 interferon ω (with 6 pseudogenes), and limitin, a mouse-specific interferon that resembles human interferon δ [12]. T1IFNs signal though interferon alpha receptor 1 (IFNαR1) and one of three co-receptors (IFNαR_2a_, IFNαR_2b_, or IFNαR_2c_). However, each T1IFN family member can elicit distinct signaling cascades due to their co-receptor availability, duration of binding to the receptor complex, and binding orientation [13,14]. During canonical type 1 interferon (T1IFN) signaling, different combinations of signal transducer and activator of transcription (STAT) protein dimers enact different pro-inflammatory and anti-pathogen transcriptional programs [10]. T1IFN signaling has established roles in promoting the development of SLE, and recent studies have demonstrated that T1IFN signaling can alter B cell tolerance at multiple stages of selection and maturation [1,15].

The type 2 interferon (T2IFN) family in mice and humans consists of only one member, interferon gamma (IFNγ), that binds to a single receptor complex (IFNγR) that consists of interferon gamma receptor 1 (IFNγR1) and 2 (IFNγR2) [16]. IFNγ signaling through IFNγR can be mediated by STAT1 homodimers, STAT3 homodimers, or STAT1:STAT3 heterodimers [16]. The effect of IFNγ signaling can differ vastly between different immune cells, but in general, IFNγ signaling promotes antigen presentation and pro-inflammatory activity by innate immune cells; enacts T-helper 1 immune responses; induces class-switched antibody production by B cells; and induces cytotoxic activity in antigen-specific CD8 T cells [17,18,19,20,21,22]. As described below, studies in mouse models of SLE indicated the crucial role of IFNγ signaling in B cells in the development of autoreactive B cells, autoantibodies, and SLE pathogenesis.

The type 3 interferon (T3IFN) family consists of four cytokines in humans, including IFN-λ1 (also called IL29), IFN-λ2 (also called IL28A), IFN-λ3 (also called IL28B), and IFN-λ4. Mice only have two T3IFNs, called IFN-λ2 and IFN-λ3 [23]. These cytokines bind to the type 3 interferon receptor which consists of the IL-10 receptor beta (IL-10Rβ, also called IL-10R2) and interferon lambda receptor 1 (IFNλR1, also called IL28Rα) This receptor primarily signals thorough STAT1:STAT2 heterodimers [23]. Several interferon lambdas (IFNλs) are produced by epithelial cells at the barrier sites, and the complete IFNλ receptor complex (IFNλR) is only expressed on a limited number of immune cells, including B cells, neutrophils, monocytes, and dendritic cells [23,24,25,26,27]. An elevated expression of T3IFN was observed in SLE patients, and the role of T3IFN signaling in autoreactive B cells remains a topic of study.

### 1.2. Peripheral B Cell Subsets and Their Roles in SLE

The peripheral B cell pool consists of immature transitional B cells and mature B cells. Transitional B cells are subdivided into transitional type 1 (T1), type 2 (T2), and type 3 (T3). Mature B cells develop from transitional cells and are broadly categorized into two distinct subsets with diverse functions: innate-like (IL-B) B cells and conventional follicular B-2 cells. IL-B cells can be further divided into marginal zone B cells (MZBs) and B-1 B cells. IL-B cells are self-renewing polyreactive B cells that generate rapid but low-affinity antibody responses. In contrast, B-2 cells are antigen-specific naïve B cells in secondary lymphoid organs that require strong B cell receptor (BCR) activation to generate a slow, high-affinity antibody response to specific antigens [28]. As described below, various interferon signaling pathways activate subsets of B cells to promote the progression of SLE.

B-1 cells are abundant in the peritoneal cavity and barrier tissues, where they can be rapidly mobilized during infection. B-1 cells express high levels of PRRs and express poorly diversified V(D)J genes that allow for a low-affinity, T-independent, and polyreactive natural IgM antibody responses during the early stages of infection. B-1 cells can also generate IgM+ memory B cells, but the bulk of long-lived antibody responses are generated via B-2 founder cells. CD5+ B-1 cells (called B-1a cells) are potentially self-reactive because they bypass the pre-BCR checkpoint of B cell development. In addition, a subset of B-1a cells can bind to apoptotic cell debris, potentially allowing them to be activated in response to self-antigens [29]. However, CD5 limits signals through the BCR, so these cells are only activated during a strong BCR stimulus [29,30].

MZBs are localized at the splenic interface between circulating blood and the lymphatic system. Like B-1 cells, MZBs carry a polyreactive BCR, high levels of surface IgM, high levels of complement receptors (CD21 and CD35) and PRRs and are a key source of natural IgM. MZBs also transport antigens from the blood to follicular dendritic cells (FDCs) in the B cell follicle to activate antigen-specific B-2 cells [31] and helper T cells [32]. As described below, MZBs in SLE-prone mice also transport self-antigens to B cell follicles and promote autoimmune B cell responses and autoantibody production by an interferon-dependent mechanism.

B-2 cells reside in the follicles as inactive (naïve) cells, but they can be activated by cognate T cell help to become both low-affinity, short-lived antibody-producing cells and high-affinity, long-lived antibody-producing cells. During an immune response, T cells activated by antigen presenting cells (APCs) undergo co-stimulatory interactions with activated antigen-specific B-2 cells at the B cell and T cell border, allowing these B cells to initially differentiate into extrafollicular antibody forming cells (AFCs), which produce low affinity, short-lived IgM and class-switched antibodies. Later in the response, activated B-2 cells enter germinal center (GC) reactions, where they undergo rapid proliferation and somatic hypermutation of the immunoglobulin genes, promoting an affinity-based selection process [33]. The GC pathway generates long-lived plasma cells that secrete high affinity antibodies [33]. Moreover, GC B cells with BCRs that harbor moderate affinity for the target antigen can become memory B cells that establish long-term protection or recirculate in the GC to mature further for selection [34]. GC B cells with low-affinity or self-reactivity undergo apoptosis and are cleared by GC-integrated tingible body macrophages [33,35,36]. The selection and maintenance of B cells in the GC are stringently regulated to prevent the development of autoantibody-producing plasma cells and memory B cells [37,38]. Alteration in BCR and TLR signaling as well as co-stimulation in B cells can lead to breaches in both the AFC and GC tolerance checkpoints, resulting in the development of autoreactive B cells, autoantibodies, and SLE. A similar breach in tolerance during the development of B-2 precursor transitional B cells, especially T1 cells, was also previously reported in SLE-prone mice [39,40].

In SLE, self-reactive B cells can persist as long-lived plasma cells or memory B cells, and some studies suggest that most self-reactive antibody-forming cells in SLE patients are derived from non-autoreactive precursors [41,42]. In addition to the development of canonical plasma cells and memory B cells, SLE-prone mice also generate “memory-like” B cells, including age-associated/autoimmune B cells (called ABCs) [43]. ABCs differ from B-1 B cells by the lack of CD43 and CD5 expression and their absence in neonatal mice. ABCs are refractory to BCR crosslinking alone but robustly respond to activation of endosomal TLR signaling (TLR7 and TLR9). BCR crosslinking with TLR stimulation can increase proliferation of these cells [44]. As described below, ABCs and other memory-like B cell subsets rapidly accumulate in SLE-prone mice and produce cytokines that further exacerbate existing autoimmunity. Furthermore, ABC-like B cells named DN2 cells have recently been described in patients with severe SLE [45].

In a typical immune response, a separation of labor occurs between the innate-like B cell subsets and the B-2 B cell subsets. Innate-like B cells are activated without T cell help and are responsive predominantly to TLR stimulation, whereas B-2 B cells require T cell help to generate highly specific antibody responses. In SLE, interferon signaling activates both groups of B cells, and promotes crosstalk between T-independent and T-dependent B cell responses.

## 2. Type 1 Interferon Signaling in B Cells and SLE

### 2.1. Role of T1IFN Signaling in B-1a Cell Responses in SLE

Although previous studies described a strong correlation between B-1a cell expansion and lupus pathogenesis in murine SLE models [46], the role of B-1a cells in disease development remains incompletely defined. B lymphoid tyrosine kinase (Blk) knockout mice, which exhibit defective BCR signaling, have increased numbers of splenic B-1a cells, which differentiate into CD138+ IgG-secreting B1-a cells that produce anti-dsDNA IgG antibodies [47]. This study further showed infiltration of B-1a cells in the kidneys of Blk knock-out mice and infiltration of B1-like cells in the tubulointerstitial space of human kidney biopsies from SLE patients [47]. However, despite these findings, the relative contribution of B-1a cells to SLE pathogenesis in this study compared to other subsets of B cells remains unclear. In addition, although it has been established that B-1a cells require type 1 interferon for their migration to mediastinal lymph nodes and predominantly produce IgM antibodies during the early stage of influenza infection [48], the role of T1IFN signaling in B-1a cell activation and expansion in SLE is unknown.

### 2.2. Role of T1IFN Signaling in MZB Cell Responses in SLE

Previous studies have highlighted the role of MZB cells and their regulation by T1IFNs in SLE mouse models. Elevated levels of circulating T1IFNs have also been shown to promote the activation and accumulation of plasmacytoid dendritic cells (pDCs) in several mouse models of SLE, including the Wiskott–Aldrich syndrome (WAS) chimera model of B cell-driven autoimmunity [49], B6.Nba2 [50], B6.Sle1.Sle2.Sle3 [51], and MRL/lpr models [52]. Subsequently, T1IFN production by activated pDCs in the MZ compartment of BXD2 lupus-prone mice was shown to deplete MARCO^+^ MZ macrophages and allow for the migration of MZB cells into the B cell follicle [53]. These findings, together with studies by the Marshak lab that showed B cell activation by immune complexes containing nucleic acids [54], suggest an interplay of BCR, TLR, and T1IFN signaling in the activation of MZB cells in SLE-prone mice. MZB cells activated by such a synergized signaling can then migrate to the follicle by a S1pr1-dependent mechanism and potentially activate follicular B2 cells and transitional B cells in the spleen by providing self-antigens [39,51,53]. Indeed, activated MZB cells were shown to transport self-antigen to the follicles and promote elevated spontaneous GC and autoantibody responses in SLE-prone BXD2 mice in an IFNαR signaling dependent mechanism [53].

### 2.3. Role of T1IFN Signaling in Regulating AFC and GC Responses in SLE

Whereas pathogen-induced AFCs and GCs generate antibodies against pathogens, SLE patients and murine models of SLE develop spontaneous GCs and AFCs that produce disease-causing autoantibodies [38,55,56,57,58,59,60]. Although the detailed mechanisms involved in the development of spontaneous GCs and AFCs in SLE are yet to be fully defined, several studies have highlighted the B cell-intrinsic requirement of T1IFN signaling in the regulation of spontaneous GC and AFC responses and increased titers of anti-nuclear antibodies in autoimmune-prone B6.Nba2, B6.Sle1b, BXD2, and bm12 cGVHD mice [53,61,62]. However, unlike in BXD2 mice, the marginal zone compartment, including MZB cells and MZ macrophages, remained intact in B6.Nba2 and B6.Sle1b mice [50,53,61,62]. Interestingly, B6.Sle1b mice overexpressing TLR7 had a reduced number of MZB cells but heightened autoimmune AFC, GC, and autoantibody responses similar to those observed in BXD2 mice [63], and these heightened responses were strongly correlated with increased T1IFN responses (63). These data indicate a differential regulation of MZ and GC B cells by T1IFN signaling. By conditionally deleting the *Ifnar1* gene in GC B cells using the Igh-γ1 Cre (Cγ1-Cre) system in B6.Sle1b mice, we also delineated a GC B cell-specific role of T1IFN signaling in promoting the spontaneous GC response and loss of tolerance [62]. We further demonstrated that T1IFN signaling promoted autoimmune B cell development in the GC pathway by regulating BCR signaling [62]. Enhanced BCR signaling was also observed in CD27+CD43+ plasmablasts from SLE patients after pre-treatment with T1IFN in vitro [64]. Furthermore, T1IFN signaling in B cells increased BCL-2 expression [50], proliferation, antigen presentation, and CD40 expression [61]. Collectively, these data suggest the involvement of T1IFN signaling in B cells in regulating B cell selection and survival within the AFC and GC pathways in SLE (Figure 1). 

In addition to the role of T1IFNs in controlling spontaneous GC, AFC, and autoantibody responses at secondary lymphoid organs, T1IFN signaling was shown to induce CXCL13 production by fibroblasts to generate ectopic GCs during the immune response to influenza infection, and the combination of IFNβ and STING signaling recruited B cells to the lung in the absence of influenza virus [65]. Although lupus-prone mice can generate ectopic GCs [66], the role of T1IFN signaling in the development of ectopic GCs in SLE has not yet been reported.

### 2.4. Model Dependent Role of T1IFN Signaling in B Cell Responses in SLE Mouse Models

Although a role for T1IFNs in SLE patients and in murine models of SLE is well established, several groups recently identified B cell-intrinsic roles for T1IFN signaling in autoimmune B cell responses and disease development that varied depending on the mouse model studied [62]. By deleting IFNαR1 in GC B cells in B6.Sle1b mice, we identified the GC B cell-specific role of T1IFN signaling in promoting the spontaneous GC, AFC, and autoantibody responses [62]. B cell responses, including GC and plasma cell responses and autoantibody production, were moderately reduced in the B6.Nba2 model after conditional deletion of IFNαR in total B cells using the Mb1-Cre mice. However, the deletion of IFNαR on B cells in B6.Nba2 mice did not affect immune complex deposition in the kidney or the development of glomerulornephritis [50]. Using the Wiskott–Aldrich syndrome (WAS) chimera model of B cell-driven SLE autoimmunity, Jackson et al. found a minor role of T1IFN signaling in B cells in autoimmune B cell responses, autoantibody production, and nephritis [49]. Consistent with the B6.Nba2 and WAS models, T1IFN signaling was also not required for TLR7-driven nephritis in NZM2328 lupus-prone mice [67], although we observed a moderate reduction in TLR7-mediated autoimmune B cell responses, immune complex deposition, and nephritis in B6.Sle1b mice [63].

In addition, the Mountz group highlighted the role of endogenous IFNβ expression in transitional type 1 (T1) B cell survival and development in BXD2 lupus-prone mice and circulating B cells in SLE patients [39,68]. The role of T1IFN signaling in the migration of MZB cells to the follicle and the concurrent disappearance of MZ macrophages was also previously reported in BXD2 mice [53], although a B cell intrinsic role of such signaling in the depletion of MZ cell populations remains to be determined. Recently, Soni et al. used Dnase1l3^-/-^ mice and identified a role for T1IFN signaling and T1IFN-produced by pDCs in the development of anti-dsDNA antibody responses and SLE-like disease observed in these mice [69]. Interestingly, in this model, T1IFN signaling promoted dsDNA-reactive B cell differentiation by the extrafollicular AFC pathway independent of the GC response. Furthermore, in SLE patients, pDC depletion confers the greatest clinical benefit to patients with a higher baseline T1IFN signature, although this study did not directly assess self-reactive B cell responses [70]. Altogether, these findings indicate a model dependent, B cell-intrinsic mechanism by which T1IFN signaling promotes autoimmune B cell responses and autoantibody production via the GC and/or the AFC pathway, influenced by additional genetic and environmental factors; however, global or B cell-specific T1IFN signaling plays a minor to moderate role in nephritis development. These studies in mouse models and data from completed clinical trials [70,71,72] further caution against a one-size-fits-all approach of anti-IFNαR blocking therapy in human SLE patients.

### 2.5. T1IFN Signaling and B Cell Responses in Type 1 Interferonopathies

Type 1 interferon signaling is controlled by several pro-inflammatory and regulatory pathways, and genetic mutations in these signaling intermediates are associated with a class of mendelian diseases called type 1 interferonopathies. Mutations in nucleic acid sensing receptors, DNA repair proteins, complement pathway intermediates, and proteosome regulators are all associated with this class of inflammatory diseases. Many type 1 interferonopathy patients develop a SLE-like disease with a single gene mutation, whereas SLE is classically defined as a polygenic disease that is largely activated by environmental triggers. SLE-like autoantibody profiles are also present in patients with type 1 interferonopathies, but our understanding of B cell-specific phenotypes in these patient cohorts is still limited. Several mouse models of interferonopathies, including *Dnase1l3*- and *Trex1*-deficient mice, and *Ifih1* mutant mice, also have elevated titers of autoantibodies [69]. Therefore, further study of B cell phenotypes in these mouse models could be used to determine how B cell tolerance may be broken during the progression of interferonopathy and whether B cells and B cell-produced autoantibodies contribute to SLE-like disease in these mice.

## 3. Type 2 Interferon Signaling in B Cells and SLE

### 3.1. Type 2 Interferon (T2IFN) Signaling in Regulating GC and AFC Responses in SLE

Whereas global or B cell-intrinsic T1IFN signaling has minor to moderate model-dependent effects on the regulation of autoimmune responses and the development of disease, targeting T2IFN (IFNγR) signaling ameliorated autoimmune-associated lymphadenopathy and autoantibody production, and provided robust protection against the development of end-stage organ disease in SLE-prone MRL/lpr mice [73]. In human SLE, a spike in IFNγ precedes the onset of SLE by several years, supporting the crucial role of this cytokine in the initial loss of tolerance [74]. In line with this report, several studies have demonstrated cell-intrinsic requirements for T2IFN (IFNγ) signaling in lupus-associated GC, AFC, and autoantibody responses and the development of nephritis in various mouse models. Lee et al. found a significantly elevated number of IFNγ-producing T cells in SLE-prone Roquin^san/san^ (sanroque) mice [75]. This group focused on the T cell-intrinsic role of excess IFNγ in autoimmunity development and showed that increased IFNγ signaling caused Bcl-6 overexpression in T cells, leading to an accumulation of Tfh cells, GC B cells, and autoantibodies and development of lupus nephritis in Roquin^san/san^ mice (Figure 2). 

Consistent with findings from the sanroque model, Jackson et al. found increased IFNγ production in the WAS chimera model of B cell-mediated SLE [49]. These authors, however, focused on the B cell-intrinsic role of IFNγ signaling through B cell-specific deletion of IFNγR in the WAS B cell chimera model and found that increased IFNγ signaling induced overexpression of *bcl6* in B cells, promoting autoimmune GC, Tfh, and autoantibody responses and the development of immune-complex nephritis. Concurrent with the study by Jackson et al., we showed the requirement of B cell-intrinsic IFNγ signaling in spontaneous GC, AFC, Tfh, and antibody responses in both non-autoimmune B6 and autoimmune-prone B6.Sle1b mice [76]. In this study, we demonstrated a B cell-specific role for STAT1 and T-bet downstream of IFNγ signaling in mediating such responses (Figure 2). In a separate study, we conditionally deleted IFNγR in B cells in a TLR7-induced B6.Sle1b SLE disease model and found that B cell-intrinsic IFNγ signaling was necessary and sufficient for promoting splenomegaly, elevated GC, AFC, Tfh, and autoantibody responses, and for the development of lupus nephritis [63].

Optimal STAT1-mediated immune responses require phosphorylation of both tyrosine 701 (STAT1-pY701) and serine 727 (STAT1-pS727). By crossing autoimmune-prone B6.Sle1b and SLE disease-prone B6.Sle1b.Yaa mice to the B6.STAT1-S727A mutant strain [77], in which serine 727 in STAT1 is replaced with alanine, we recently discovered the role of STAT1-pS727 in promoting systemic autoimmunity and SLE disease by regulating AFC, GC, Tfh, and autoantibody responses [78]. By generating B cell-specific BM chimeras, we further demonstrated a B cell-intrinsic role for STAT1-pS727 in autoimmune AFC, GC, and autoantibody responses. However, we found no contribution of STAT1-pS727 to foreign antigen- or pathogen-driven GC, Tfh, and Ab responses. STAT1-pS727 also did not play a significant role in GC and Tfh responses in gut associated lymphoid tissue (GALT) in B6.Sle1b mice, responses that were previously shown to be mediated by gut microbiota and dietary antigens [79,80,81]. Collectively, our data indicate a differential regulation of autoimmune and pathogen driven responses by STAT1-pS727 (Figure 2). Thus, STAT1-pS727 downstream of TLR- and IFN-signaling is an intriguing candidate for the implementation of targeted SLE therapeutics that preserve anti-microbial immunity.

### 3.2. T2IFN Signaling and Age-Associated B Cells (ABCs)

CD11c^+^T-bet^hi^ age-associated B cells (ABCs) play an important role in autoimmunity development and are expanded in several autoimmune diseases including SLE [43,82,83]. TLR (TLR7 or TLR9), IFNγ, and IL-21 signaling are crucial for the development and maintenance of ABCs in both murine models of SLE and SLE patients [43,84]. Although BCR signaling was not required for the generation of ABCs, TLR signaling (via MyD88) in coordination with BCR activation increased ABC activation, cytokine production, and survival [44,84]. In turn, activated ABCs produce TNF-alpha, IFNγ, and anti-nuclear antibodies that contribute to SLE development [84,85], and targeted depletion of ABCs in mice with dampened autoantibody production [84]. Recently, several groups identified CD11c^+^T-bet^+^ ABCs in SLE patients, including the accumulation of TLR7-driven and T-bet-expressing IgD^neg^CD27^neg^CD11c^+^CXCR5^neg^ (named DN2) pre-antibody secreting cells (pre-ASCs) [45]. Consistent with studies in mice, Lund and Colleagues showed a strong correlation between elevated levels of IFNγ in SLE patients and expansion of DN2 cells [86]. They further showed that IFNγ induced epigenetic programming of T-bet^hi^ human DN2 cells to promote TLR7/TLR8 and IL-21 driven differentiation [86]. At present, several groups are exploring the potential of ABC-targeted therapeutics as an approach to disrupt IFNγ and autoantibody production and SLE progression.

### 3.3. T1IFN and T2IFN Signaling in the Regulation of B Effector Cell Subset

Another subset of cytokine-producing B cells, called B effector cells (Be1), was previously reported to be regulated by both T1IFN and T2IFN signaling. T1IFN and T2IFN signaling in B cells was shown to coordinate the generation of Be1 cells from naïve and memory B cell precursors [87]. After activation with TLR and CD40 signaling, Be1 cells produced IFNγ and IL-12 to stimulate further T helper 1 responses in secondary lymphoid organs [88] (Figure 2). We also previously showed the production of IFNγ and expression of T-bet by autoimmune spontaneous GC B cells, but not by foreign-antigen induced GC B cells [76], suggesting that spontaneous GC B cells acquire a Be1 cell-like phenotype during autoimmune responses. The combined activation of Be1 cells by T1IFN and T2IFN signaling was shown to be mediated by a sequential activation of STAT1 and STAT4 to induce T-bet expression [87]. Although we and others showed a crucial role for STAT1 in IFNγ signaling and autoimmunity [76], we found STAT4 to be dispensable in the formation of spontaneous GCs, T cell activation, complement deposition in the kidney, or the generation of T helper 1 cells in various autoimmune- and SLE-prone mice [89].

### 3.4. T1IFN and T2IFN Signaling in Regulatory B Cells

While IFNγ signaling induces a pro-inflammatory type 1 immune response, it concurrently negatively regulates immunoregulatory pathways, including inhibition of regulatory B cell responses. Regulatory B cells (Bregs) are a subset of B-2 cells that secrete IL-10, IL-35, and TGFβ to inhibit inflammatory responses [90,91,92,93,94]. In humans, pDCs isolated from healthy patients can induce Breg differentiation from immature B cell precursors by a T1IFN- and CD40-dependent mechanism, and in turn, these Bregs secrete IL-10 to regulate pDC activation and several other immune pathways [95]. However, in co-culture studies, pDCs from SLE patients failed to promote Breg expansion. We recently showed elevated expression of TLR7, STAT1, and IFNγR in CD19^+^CD5^+^CD1d^+^ regulatory B cells in mice [96]. We demonstrated that increased TLR7 activity promoted type 1 B cell responses including GC, AFC, and IgG2c antibody responses in autoimmune-prone mice but also inhibited the generation of Bregs, predominantly in an IFNγ signaling-dependent manner [96]. Interestingly, in human B cells, IFNγ inhibits the generation of IL-10 by primary B cells after treatment of these cells with agonists for TLRs 2,3, 4, 7, or 8, but IL-10 production is increased by IFNγ after B cells were treated with a TLR9 agonist [97], consistent with an established negative regulatory role of TLR9 in autoimmune B cell responses [98,99,100,101]. After combined IFNγ and TLR9 agonist treatment, a co-culture of treated B cells with T cells also inhibited T cell proliferation in vitro [97]. These data indicate that T1IFN and T2IFN signaling differentially regulates type 1 autoimmune and Breg responses in TLR7-mediated systemic autoimmunity.

## 4. Type 3 Interferon Signaling in SLE

An elevated expression of interferon lambda (IFNλ) has been associated with several autoimmune diseases, including SLE, rheumatoid arthritis, Sjogren’s syndrome, and systemic sclerosis [102,103,104,105,106,107]. In SLE patients, serum IFNλ concentration positively correlated with SLE disease activity index (SLEDAI) scores, titers of anti-nuclear antibodies, and complement activation [102,103,108,109,110]. IFNλ was expressed in barrier tissues, making it a crucial mediator of SLE-associated skin inflammation. Surprisingly, IFNλ was also involved in autoimmune inflammation at non-barrier sites, including the joints and kidneys [102,108,111]. These data suggest a potential role for T3IFN signaling in the development of B cell-driven autoimmunity, although the direct role of T3IFN signaling in autoimmune B cell responses in SLE remains poorly understood.

### 4.1. T3IFN in Human B Cell Activation

IFNλR1 expression was shown to be upregulated on human naïve and memory B cells [112], and stimulation of naïve B cells with BCR crosslinking and interferon lambda induced rapid differentiation of plasmablasts that produced IL-6 and IL-10 [113]. When combined with BCR crosslinking, IFNλ increased BCR signaling through the mTORC1 pathway to induce cell proliferation [113]. IFNλ along with TLR7/8 stimulation of human B cells increased the expression of the early activation marker CD69 and cytokines (IL-6 and IL-10), antibody production, and proliferation [112]. The expression of IFNλR on CD27^+^ memory B cells and the synergistic effects of IFNλ and TLR stimulation on B cell activation suggest that T3IFN signaling could play a crucial role in regulating ABC or DN2 cell activity in SLE, although a link between IFNλ signaling and ABCs or DN2s has not been established.

### 4.2. T3IFN Signaling in Autoimmune B Cell Responses in SLE-Prone Mice

Goel et al. showed that IFNλ was elevated after epicutaneous treatment of a TLR7 agonist, and broadly promoted systemic autoimmune responses as indicated by splenomegaly, increased white blood cells, decreased hemoglobin and platelets, myeloid expansion, neutrophil NET production, and T cell activation [114]. Surprisingly, IFNλ had no direct role in B cell activation, IgG+ plasma cell development, or autoantibody production, although these mice exhibited a modest reduction in the proportion of naïve B cells in the spleen [114]. However, during infection of mice with blood-stage malaria, B cell intrinsic IFNλR expression protected mice from severe disease by increasing CD138+ plasma cells and parasite-specific antibodies [115], indicating a clear role of IFNλR signaling in modulating B cell activity in mice during infection. Moreover, contrary to SLE-prone mice, IFNλR signaling induced a robust activation of interferon-stimulated genes in B cells from SLE patients [114]. This discrepancy in IFNλ-mediated B cell activation in human SLE patients versus TLR7-induced mice is currently not clear. Further studies of the B cell-intrinsic role of T3IFN signaling in autoimmune B cell responses in various SLE mouse models are warranted to reach a consensus regarding the role of IFNλ in B cell activation in mice.

## 5. Concluding Remarks

SLE is a complex B cell-mediated autoimmune disease caused by genetic and environmental factors. Here, we have discussed the roles of all three classes of interferons (types 1, 2, and 3) in the regulation of autoimmune B cell responses, production of autoantibodies, and development of lupus nephritis. Existing literature from various autoimmune- and SLE-prone mouse models suggests a minor to moderate model-dependent B cell-intrinsic role of T1IFN signaling in autoimmune B cell responses and development of lupus nephritis [116,117,118,119,120,121]. T2IFN signaling in B cells, on the other hand, plays an indispensable role in promoting autoimmune B cell responses and the development of SLE. The role of T3IFN signaling in B cell responses in SLE is incompletely understood and further investigations of B cell-intrinsic T3IFN signaling in autoimmune B cell responses in various mouse models is warranted. Altogether, the published data suggest a differential regulation of autoimmune B cell responses in SLE by the three classes of interferons. A focus on the genetic and environmental factors that contribute to the differential regulation of autoimmune B cell responses by interferons could be vital for the development of improved interferon-targeted treatments for SLE.

## Figures and Tables

**Figure 1 ijms-22-10464-f001:**
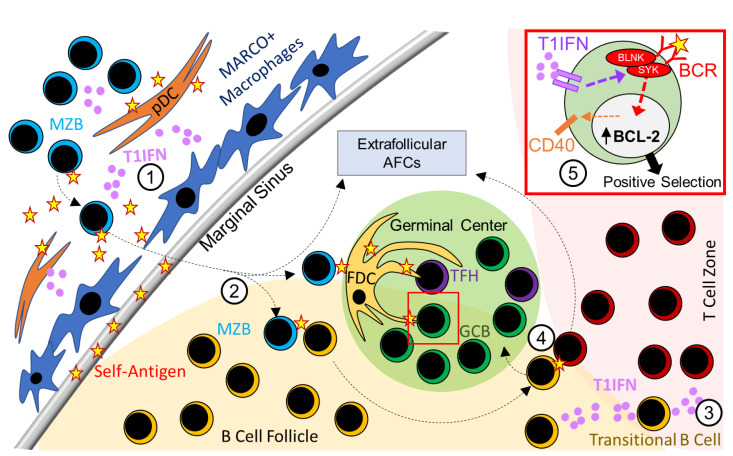
Type 1 interferon (T1IFN) signaling promotes autoreactive B cell generation in the extrafollicular AFC and follicular GC pathways. (1) T1IFN is predominantly produced by activated plasmacytoid dendritic cells (pDCs) at the splenic marginal zone. (2) Elevated T1IFN promotes the removal of MARCO1+ macrophages and allows for the migration of marginal zone B cells (MZBs) that carry self-antigen to the B cell follicle to activate self-reactive B cells. (3) Transitional B cells in the T cell zone can also produce T1IFN to further activate naïve B cells and T cells. (4) After acquiring self-antigens from MZBs, activated follicular B cells interact with cognate T cells, and differentiate into extrafollicular antibody-forming cells (AFCs) or mature through germinal centers. (5) Within the germinal center, T1IFN signaling in GC B cells augments BCR signaling, and promotes proliferation, CD40 expression, and BCL-2 expression to promote the survival and positive selection of self-reactive B cells.

**Figure 2 ijms-22-10464-f002:**
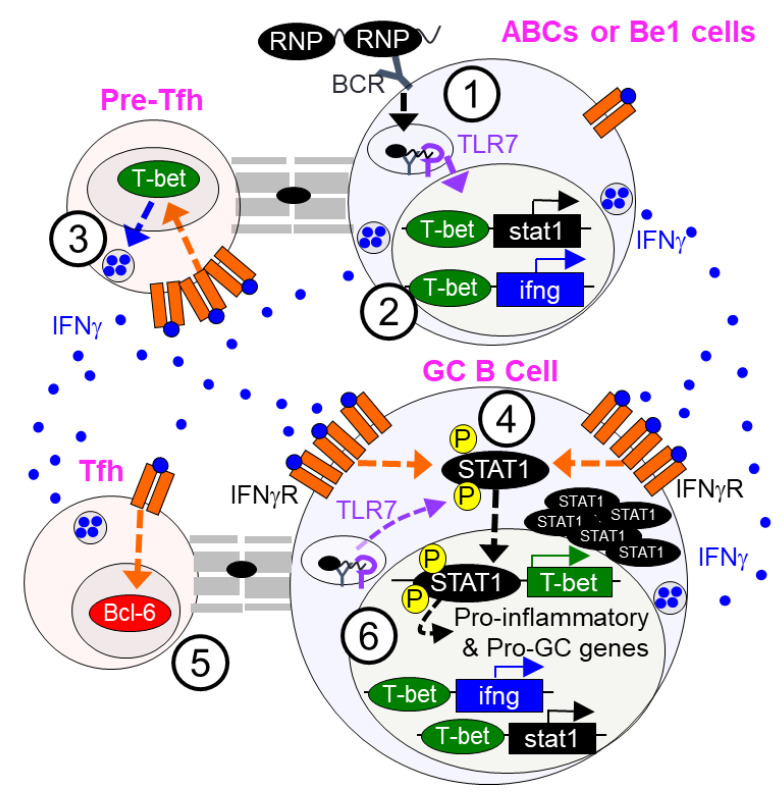
**Type 2 interferon signaling in the germinal center.** (1) B cell receptor cross-linking by self-antigen immune complexes activates TLR7 to induce (2) Tbet-mediated production of IFNγ in age-associated B cells (ABCs) and B effector 1 (Be1) cells. (3) Cognate interaction of B cells with T cells outside of the germinal center allows pre-T follicular helper (pre-Tfh) cells (CD4^+^CXCR5^Int^ PD1^Int^) to produce IFNγ. (4) Within the GC, Tfh-derived IFNγ signals in B cells to induce the phosphorylation of STAT1 (at Y701) to promote Tbet transcription and further IFNγ production. (5) In fully differentiated GC-Tfh (CD4^+^CXCR5^Hi^PD1^Hi^) cells, IFNγ signaling increases BCL6 production and promotes GC stability. (6) Finally, combined TLR7, BCR, and IFNγ signaling in GC B cells promotes the phosphorylation of STAT1 at two phosphorylation sites (Tyr 701 and Ser 727), allowing for the full activation of STAT1-mediated GC stability and pro-inflammatory gene programs.

**Table 1 ijms-22-10464-t001:** Classification of type 1, 2, and 3 interferon members and interferon receptors in humans and mice.

	Human		Mouse	
Interferon Family	Receptor	Group Members	Receptor	Group Members
Type 1	Primary: IFNαR1 Co-receptors: IFNαR2a IFNαR2b IFNαR2c	14 Alphas (α), (2 pseudogenes) 1 Beta (β) 1 Omega (ω) 1 Delta (δ) 1 Nu (ν), (pseudogene) 1 Epsilon (ε)	Primary: IFNαR1 Co-receptors: IFNαR2a IFNαR2b IFNαR2c	13 Alphas (α), (1 pseudogene) 1 Beta (β) 7 Omegas (ω), (6 pseudogenes) 1 Kappa (κ) 1 Epsilon (ε) 1 Limitin (similar to interferon δ)
Type 2	IFNγR1 IFNγR2	1 Gamma (γ)	IFNγR1 IFNγR2	1 Gamma (γ)
Type 3	IL28Rα IL-10Rβ	IL-29 (IFNλ1) IL-28A (IFNλ2) IL-28B (IFNλ3) IFNλ4	IL28Rα IL-10Rβ	IFNλ2 IFNλ3

## Data Availability

Not applicable.

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
