# Peer review of "Regulation of B Cell Responses in SLE by Three Classes of Interferons"

_ijms, 2021, doi:10.3390/ijms221910464_

Round 1

Reviewer 1 Report

In this review manuscript the authors give an excellent and concise overview of the regulatory role of three classes of IFN signaling in SLE B cells. The manuscript is well written. The following are minor issues. 

Minor issues

  1. Page 1, line 24, “ ….. by autoantibodies that target immune responses to specific organ systems”. Remove “target immune responses to”
  2. Page 2, line 64, “T1IFN signaling has established roles in promoting the development of SLE, but recent studies have demonstrated…” change “ but” to “and”.

Author Response

We would like to thank the reviewer for suggested corrections. We revised both sentences as suggested in the revised manuscript.

Reviewer 2 Report

This is an up-to-date review on a complex topic.
The work is not always easy to read as data form human SLE and murine models of the disease are often considered together.
Conversely, there is no focus on the monogenic forms of SLE in humans and on the role of Interferons in autoimmunity in the emerging group of interferonopathies, which may share with SLE significant pathogenic mechanisms. 

Author Response

We would like to thank the reviewer for insightful comments on this review. We agree that this topic is complex and continuously evolving. We combined information about human SLE and mouse models to reduce redundancy in the manuscript. However, we revised some parts of this manuscript for clarity.

As suggested by the reviewer, we also included a paragraph with an introduction to interferonopathies on lines 288 to 304 of this manuscript. We focused this review on highlighting the roles of various interferon signaling in the development of autoreactive B cells in SLE. Specific studies of B cells in patient populations or mouse models of Type 1 interferonopathies are limited, but we agree that this class of diseases should be highlighted as a part of this review.   

Reviewer 3 Report

  1. The title may need rephrasing.
  2. The sentences in Introduction end abruptly. Ensure  the cascade flow of information and uniformity. 
  3. Recheck information mentioned in Line #57
  4. Lines 75-76 need rephrasing.

Author Response

We would like to thank the reviewer for insightful suggestions on how to improve this manuscript. As suggested by the reviewer, we have revised the title for clarity. Our revised title is “Regulation of B cell responses in SLE by three classes of interferons”. We also revised portions of the introduction to improve the readability. We also made corrections to the manuscript as suggested in comments 3 and 4.

Round 2

Reviewer 2 Report

The manuscript has been improved to address the raised issues